# The impact of national culture, altruism, and risk preference on salaries: The case of the Major League Baseball

**Wen-Jhan Jane[1], Yi-Jie Yu[1], Jye-Shyan Wang** [2,3]*

**1** Department of Economics, Shih Hsin University, Taipei, Taiwan, **2** Department of Physical Education and Sport Sciences, National Taiwan Normal University, Taipei, Taiwan, **3** Department of Athletics, National Taiwan University, Taipei, Taiwan

* jyeshyan@ntnu.edu.tw

**Data Availability Statement:** All relevant data are within the paper and its Supporting Information files.

**Funding:** The author(s) received no specific funding for this work.

## Abstract

Based on the longitudinal data of 30 Major League Baseball (MLB) teams over seasons from 2017 to 2020, we used random effect (RE) models to conduct regression analyses on the detailed data of pitchers and fielders. Cultural distance (CD) was measured in terms of Hofstede's cultural indicators and Global Preference Survey (GPS) data. The results showed that salary premiums for foreign MLB players existed and CD was significantly positively correlated with salaries. Further, the risk preference (/altruism) difference between foreign pitchers and American pitchers was significantly positively (/negatively) correlated with the salaries of foreign pitchers. Salary estimation data showed that the salary premium was nearly 20% for players from South Korea and Panama, the lowest (only 0.11%) for players from Australia, and only 6.13% for players from Dominican Republic (accounting for the largest proportion of foreign MLB players), indicating that the MLB's foreign player recruitment policy is correct.

## 1 Introduction

The first professional baseball game in the world was played in the USA in 1869 when the Cincinnati Red Stockings and the Mansfield Independents opened a new era of professional baseball. Today, the MLB with 30 baseball teams has received much attention in professional sports worldwide. In addition to the US, many Asian countries also have their own professional baseball organizations, including Japan's Nippon Professional Baseball (NPB), South Korea's Korea Baseball Organization (KBO), and Taiwan's Chinese Professional Baseball League (CPBL). Hence, the baseball sport has developed to a mature level. While the sports industry is increasingly internationalized, professional sports leagues have successively recruited foreign players to increase the competitiveness of their teams and attract viewers. The MLB employs talented baseball players worldwide with different nationalities. The salary gap between such players has become a hot topic of research in labor economics.

Salary discrimination is a very important topic in labor economics. The term "salary discrimination," which was first proposed by Gary Becker in *The Economics of Discrimination*,

**Competing interests:** The authors have declared that no competing interests exist.

refers to how people are treated differently based on racial origin, gender, personal characteristics, and age, as well as prejudice against certain specific ethnic groups. Most studies of salary discrimination in professional sports focus on players' nationality, racial origin, salaries, personal characteristics, and performance. Such studies first examined the salary gap between players from different races, and then extended to the salary gap between players with different nationalities. Jane, Chen, and Kuo; Swift; and Wang, Fang, and Wu respectively examined the salary gap between players of the NPB, US' Major League Soccer (MLS) and KBO, finding that nationality indeed has a significant impact on players' salaries [1–3]. In summary, the salary gap between players is mainly due to two reasons. Foreign players are paid more than domestic players, probably due to lack of information about foreign players and overestimation of their performance; foreign players are paid less than domestic players because high uncertainty leads to a relatively conservative attitude toward their salaries [1].

To increase the win rate and improve the professional ranking and prestige, professional sports teams of many countries actively recruit international players in addition to domestic players. To protect the rights and interests of domestic players, the NPB, KBO and CPBL, which represent the highest level of professional baseball in Asia, have established a so-called "foreign player system" to limit the number of foreign players. By contrast, the MLB has not established such regulations. According to the MLB's player roster of 2017 to 2020, foreign players account for 25%, playing a due role in the MLB. Among foreign players, as presented in the Fig 1, 243 players from Dominican Republic(DO) account for 42%, sequentially followed by players from Venezuela (VE), Cuba (CU), and Mexico (MX); together these four countries account for nearly 82.5% of the foreign players. Further, 15 foreign players are from Colombia (CO) and 15 are from Japan (JP); 71 foreign players are from other countries, accounting for 12%. Evidently, the MLB is a highly internationalized professional sports league.

Fig 2 shows the salary gap by nationality, or specifically, salary gap between players from the top six countries. Players from JP enjoy the highest average annual salary ($8,323,600), which is approximately 2.79 times that of US players, and approximately 1.63 times that of Cuba players. Compared with US players, players from these countries are considered to fall within minorities, but are paid significantly more; this conflicts with the traditional literature on salary discrimination against minority groups. This may be due to the information asymmetry between foreign players and their American employers, making it impossible to accurately determine reasonable salaries for foreign players in the process of salary bargaining; therefore, the baseball teams pay foreign players higher salaries, thus resulting in inverse salary discrimination and salary gap between players by nationality [1]. In addition, the average annual salary of players from VE, MX, DO, and CO is $ 77,000 to $ 237,300 lower than that of their American counterparts.

In most of the previous studies of salary discrimination in professional sports, the salaries of players were assessed in terms of racial origin or nationality, but rarely from a cross-cultural perspective. Each country has its own specific background, belief, and cultural values, suggesting that the cultural difference (CD) between countries has a very significant impact on salaries. Therefore, this study examined the impact on salaries of a professional sports league from the perspective of CD. The theory of Hofstede, an important expert in international cultural studies, has been widely used in the study of international communication and international corporate relationships. Using cultural indicators proposed by Hofstede, we analyzed the highly internationalized professional sports industry to reveal the impact of players' cultural background on their salaries. In this study, CD is classified into six cultural dimensions, including the power distance index (PDI), individualism or collectivism (IDV/COL), uncertainty avoidance index (UAI), masculinity or femininity (MAS/FEM), long-term orientation or short-term orientation (LTO/STO), and indulgence versus restraint (IVR) [4].

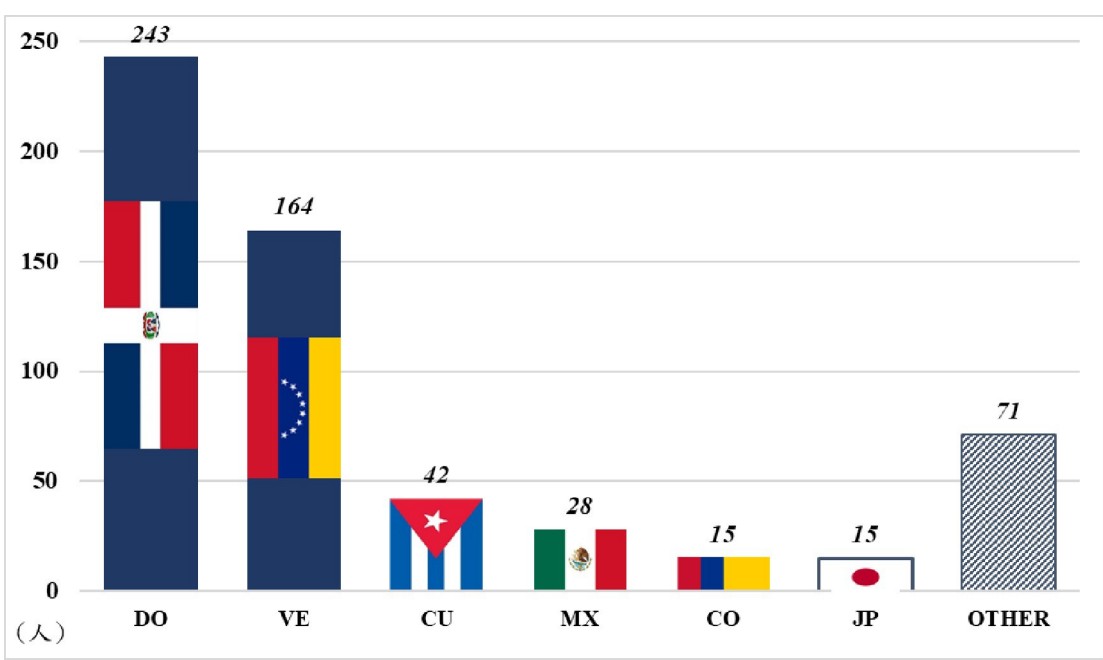

**Fig 1. Nationality distribution of foreign MLB players from 2017 to 2020.** Data source: https://www.baseball-reference.com/.

Regarding the impact of CD on salaries, Jane analyzed the data of Japan's professional baseball, finding that CD is positively correlated with salaries; specifically, the average salary of foreign players is 0.74% to 0.95% higher than that of Japan players [5]. In addition, studies showed that employee salaries are not only measured in terms of objective performance, but are also affected by subjective performance evaluations; for example, 34% of employees in the

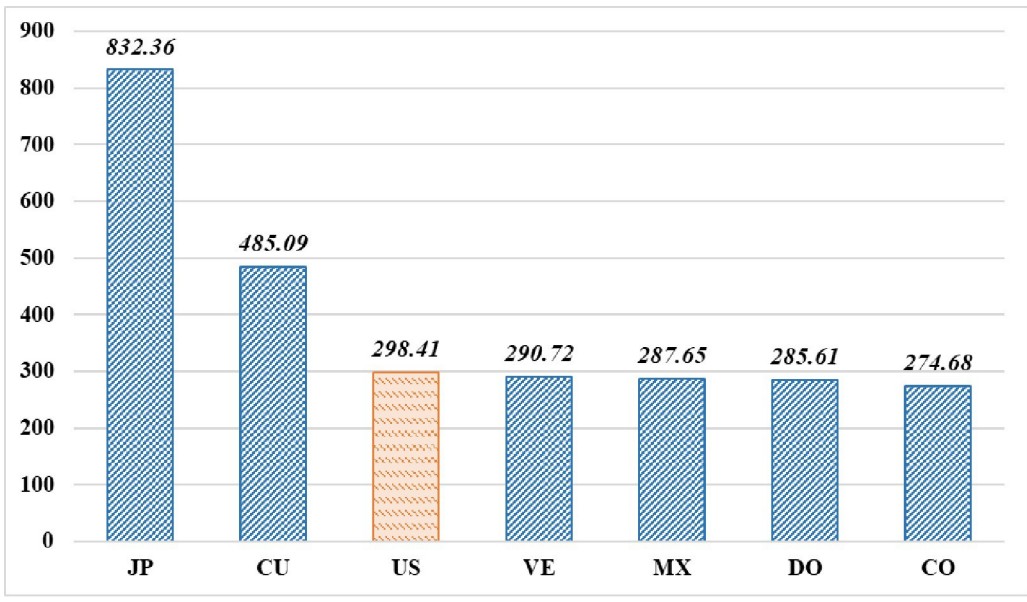

**Fig 2. Average annual salary for the MLB's foreign players and American players (unit: $ 10,000).** Data source: https://www.baseball-reference.com.

UK's industrial sector earn a salary depending on the subjective judgments of their supervisors or managers on individual performance [6, 7]; employees can not only receive a bonus, but also maintain the recruitment relationship if the subjective performance evaluation is high. Personal preferences are closely related to individual behaviors or outcomes, and personal behaviors may be influenced by risk and patience preferences. Given the accelerated pace of globalization, people's psychology and behaviors are changing over time, and cultural changes may be more intense than ever. Hence, cultural changes should be verified in a scientific way, thus preventing and solving social problems effectively. For the estimation of CD, we cited the GPS data acquired by Falk et al. [8]. Such preferences are considered as a country's potential long-term beliefs and cultures. Few studies have examined the impact of players' cross-international cultural background on their salaries using a sample of professional league sports.

This study makes an important contribution to the literature on salary discrimination from a cultural perspective, and the detailed data of professional sports can be used to estimate the impact of players' nationalities on their salaries. Using a sample of pitchers and fielders from 30 MLB teams from 2017 to 2020, this study examined the relationship between nationality, CD, and preferences of MLB players and their salaries. The salary premium for foreign players may be due to information asymmetry or a preference for foreign players. The findings of this study will help the MLB's baseball teams review their salary standards from a cultural perspective, thus developing an appropriate salary incentive policy. Among the estimation results of different cultural dimensions, the difference in collectivism and altruism between foreign and American fielders is significantly positively correlated with salaries; accordingly, foreign players with altruism and small salary gap were selected to examine how to prevent an excessive salary premium. The objective is to provide a reference for the MLB's foreign player recruitment strategy, and gain insight into salary discrimination in multinational organizations and workplaces.

## 2 Literature review

### 2.1 Salary discrimination in baseball

Previous studies of professional sports did not reach a consensus on salary discrimination. Pascal and Rapping analyzed the racial origin, salaries, and performances of MLB players, finding that racial origin had no significant impact on salaries [9]. Using a salary equation to analyze MLB players data from 1968 to 1969, Scully found that black players (outfielders) and white players were paid differently for the same work, suggesting that black players were paid less than white players while their performance was considered [10]. Thus, Scully provided evidence of salary discrimination in the MLB [10]. In addition, studies showed that teams free of racial discrimination and willing to employ black players were more likely to achieve better competition results. Such teams employed excellent players at a low salary cost, and these factors caused teams with racial discrimination to lose their competitive edge and be driven out of the market, leading to the conclusion that racial discrimination would gradually disappear [11]. Over time, studies showed a change in the evidence of salary discrimination in the MLB discrimination. Christiano analyzed the game data of 203 players in 1977 and 356 players in 1987 in terms of eight variables including the number of baseball seasons before 1987, batting average in the last year's season, number of home runs in the last year, retransmission royalty, guard position, left or right-handed batter, playoff or not, and racial origin. The results of multivariate regression analysis showed that inter-racial salary discrimination existed among the baseball teams in 1977, but the impact of racial origin on player salaries was not significant in 1987, indicating that racial origin had a different impact on player salaries ten years later. Subsequent studies also focused on inter-racial salary discrimination [12].

With the advancement in information and transportation technologies and the increasing internationalization of baseball sports, many countries are looking to recruit excellent players; in this situation, salary discrimination has extended from domestic racial discrimination to nationality-specific salary discrimination. Based on 511 observations of free agents from 1998 to 2006, Holmes defined four races including blacks, whites, Hispanic blacks, and Hispanic whites, and conducted a quantile regression (QR) analysis, finding that in the low salary group, white and Hispanic players enjoyed a salary premium compared with black players, but this did not occur in the high salary group. In addition, the results of weighted least square (WLS) showed that Hispanic players enjoyed an approximately 11% salary premium and white players enjoyed an approximately 16% salary premium [13]. Using various methods (e.g., ordinary least squares (OLS), random effect (RE) model, and two-stage random effect model), Jane, Kuo, Wang, and Fang analyzed the data of a sample of 1,737 MLB players from 2009 to 2013, finding that Latino players suffered salary discrimination [14].

## 2.2 Salary discrimination in other professional sports

In addition to the baseball sports, studies on salary discrimination have also been conducted in other sports [2, 15–20]. Regarding salary discrimination in football, Swift analyzed the data of 1,397 MLS players; the results of OLS regression analysis showed that foreign players were paid 15.97% more than American players, which was interpreted as nationality-specific salary discrimination [2]. Ducking, Groothuis, and Hill analyzed the data of 651 line backers, 708 defensive linemen, and 926 defensive back players employed by the National Football League (NFL) from 2001 to 2008, divided them into the black player group and non-black player group, and used the OLS and QR methods to analyze the two groups; the analysis results showed that the black defensive linemen and defensive back players enjoyed a salary premium, whereas the black line backers suffered salary discrimination [15]. Using the OLS and two-stage fixed-effect model (2sFE) methods, Hill and Groothuis examined NBA players in the 1989–1990 and 2012–2013 seasons, finding that foreign NBA players enjoyed a salary premium at the early stage and did not suffer salary discrimination at the later stage [16]. Simmons summarized the research findings of papers published in *Journal of Sports Economics* on the workings of professional sports labor markets. In sum, the evidence of hiring and exit discrimination in various leagues looks stronger than the evidence for salary discrimination [21].

## 2.3 Salary discrimination in Asian professional sports

Previous studies on salary discrimination involved not only professional sports in North America, but also professional sports in Asia. Studies on salary discrimination in professional sports in Asia are fewer than those in North America, and will be discussed in the following section. Jane examined salary discrimination against indigenous players of the CPBL [22]. Using the RE and 2sFE methods, Jane analyzed the data (including personal characteristics, performance, and racial origin) of 437 CPBL players from 1990 to 2007, finding that indigenous players enjoyed a salary premium compared with Taiwanese players (specifically, the average salary of indigenous players was NT$ 25,000 to NT$ 27,000 higher than that of Taiwanese players, or 20% to 22% higher), or rather, inverse salary discrimination existed. Through OLS regression analysis, Wang et al. studied the data of 775 South Korean professional baseball players from 2001 to 2010, finding that foreign pitchers were paid more than South Korean pitchers with the same performance [3]. Using the OLS and QR methods, Jane et al. analyzed the data (including salaries, personal characteristics, performance, and racial origin) of 663 Japanese professional baseball players from 2000 to 2008, finding that nationality was significantly positively correlated with player salaries, or specifically, international players

were paid more than Japanese players (i.e., inverse salary discrimination existed). This is considered to be due to asymmetric information about international players and transaction uncertainty arising from the difference in management norms between Japanese and international players [1].

## 2.4 Impact of cultural indicators on salary

In addition to the perspective of nationality or racial origin, salary discrimination can also be studied from a cross-culture perspective. By surveying more than 116,000 questionnaires from multinational enterprises from 1968 to 1972, and studying 40 countries through literature review, Hofstede summarized four cultural dimensions (including PDI, IDV or COL, UAI, and MAS or FEM) to quantify the CD between two countries, and subsequently (2011) introduced two additional indicators including LTO or STO, and IVR [23]. According to a report by the international firm Towers Perrin[1], the salary gap between CEOs and manufacturing employees was the largest (specifically, as much as 40 times or more) in Brazil, Venezuela, South Africa, Malaysia and Mexico. To explain the salary gap in different countries, Grenness studied the salaries of CEOs and employees in 25 multinational companies by using Hofstede's three cultural dimensions as explanatory variables, finding that the average salary of CEOs was the highest ($ 1.4 million) in the USA with the highest IDV score (91), and PDI was significantly positively correlated with salary gap; evidently, cultural dimensions can account for salary gap [24]. Considering the position and influence of employees with different national cultures, Merluzzi investigated the network relationship between employees of a USA company operating in different Asian countries through a questionnaire survey of new product planning by email; the results of OLS estimation showed that senior managers in collectivistic Asian countries (e.g., South Korea, Thailand, Indonesia, and Malaysia) were paid more than those in individualistic countries (e.g., Australia)[2] [25].

Studies on salary premium have been extended to the cultural background and individual levels to examine the impact of national cultures on financial and non-financial rewards more comprehensively. Chiang and Birch examined the impact of cultural background on financial and non-financial rewards among employees at different levels in 28 Finland banks and 36 Hong Kong banks from 2000 to 2003. Financial rewards include base salaries, individual bonuses, team bonuses, and intra-organizational promotions; non-financial rewards include recognition by supervisors, work-life balance, and employee training [26]. Hofstede's six cultural dimensions are used as a framework of study. The Finnish society is a relatively feminine and individualistic society with low power distance and high uncertainty avoidance, whereas the Hong Kong society is a relatively masculine and collectivistic society with high power distance and low uncertainty avoidance. The results of a questionnaire survey on employees were analyzed through multivariate analysis of variance (MANOVA), and the analysis results showed that respondents in Hong Kong and Finland presented significant differences. The incentive effect of flexible work arrangement and employee training on employee performance in Finland is more significant than that in Hong Kong. Compared with Finnish employees, base salaries and promotions have a more significant impact on collectivistic and masculine Hong Kong employees than non-financial rewards. Different cultural dimensions provide an important perspective for studying the salary gap between different groups; organizations with a good use of cultural differences can win a competitive advantage. In addition to company employees, the studies on salary premium also extend to professional sports. Based on a sample of 1,520 NPB players from 2009 to 2017, Jane used Hofstede's CD variable and diverse methods (e.g., 2sFE and RE model) to examine the impact of nationality and CD on salaries, finding that baseball teams may overestimate the salaries of foreign players. NPB employers cannot

acquire sufficient information about foreign players, and thus end up granting salary premiums to them. Compared to Japanese players, foreign batters averagely enjoy a 0.93% to 0.95% salary premium and foreign pitchers averagely enjoy a 0.74% to 0.76% salary premium, proving the prevalence of a special tenure system in the Japanese labor market [5].

In this study, we cited national structure data from the GPS to measure the impact of attitude differences between players on their salaries. Using the organizational citizenship behavior (OCB) as the framework of study, Alkahtani conducted a questionnaire survey on altruistic behaviors in Pakistan's banking sector. The OCB comprises five dimensions including altruism, courtesy, conscientiousness, civic virtue, and sportsmanship. The study aimed to ascertain whether rewards (including salary increases and promotions) are affected by these five dimensions. The results of general linear model analysis showed that altruism is significantly positively correlated with salary increases, and the value of altruism is far higher than that of the other OCB dimensions, implying that employees can be rewarded more if they voluntarily help others with organization-related duties [27]. In the context of professional sports, Asghar and Asif verified the relationship between NBA players' salaries, performance, and altruism by conducting regression analysis on a sample of 1,566 players from 30 teams during four regular seasons in 2012−2013 and 2015−2016; in their study, altruism was measured in terms of the data on teammates' passing assists. The analysis results revealed the following: 1) Salaries are significantly positively correlated with altruism (i.e., the higher the salaries employee are paid, the more likely they are to make extra efforts at work); 2) Altruism is significantly positively correlated with the performance score, and has a mediated effect on the relationship between salaries and performance. Studies of risk preferences show that the degree of risk aversion varies from individual to individual. Specifically, some individuals are risk-averse, whereas others are risk-neutral or risk-seeking [28]. Pappas and Flaherty investigated the moderating effect of individual-difference variables on salaries, finding that the mixing proportion of floating and fixed salaries vary with risk attitudes; in their study, a questionnaire survey was conducted on a sample of 1,000 American salesmen, and risk attitudes were measured using the vocational preference inventory (VPI) [29, 30]; the results showed that risk attitudes are significantly positively correlated with floating salaries. Just like performance-based rewards, floating salaries are attractive to risk-seeking individuals; by contrast, fixed salaries are characterized by higher security and lower risk, and are favored by risk-averse individuals. Moreover, other studies argued that risk-averse employees may prefer the bonus-incentive salary system [31]. Through questionnaire survey, Deckop, Merriman, and Blau measured the individual risk preferences of American employees with an average age of 30 to 40 years, finding that the lower the degree of employees' risk aversion, the higher the degree of their satisfaction with their salaries. This implies that the pay for performance (PFP) program implemented by companies should consider employees' risk preferences [32].

## 3 Study methodology and data description

Previous studies of salary discrimination in North America and Asia showed that salary discrimination indeed exists among professional sports leagues due to differences in racial origin or nationality. Cultural indicators provide a new perspective for the studies of salary discrimination in professional sports. Based on related player data (e.g., salaries, nationality, personal characteristics, and performance) and using the salary equation of Mincer and methodology of Jane, we constructed a model to examine the impact of nationality and CD on player salaries [5, 33].

$$logSal_{it} = \alpha + u_i + \beta_1\,Nat_i + \beta_2 CD_i + \beta_3 P_{it} + \beta_4 C_{it} + v_{it} + \varepsilon_{it} \qquad (1)$$

Because the data type is panel data, fixed-effect (FE) and RE models are used based on the

regression model of Jane [5]. A houseman test must be conducted first before a FE or RE model is selected. In this study, the major variable in question is nationality, which is time-invariant; hence, it is difficult to observe the impact of nationality and CD on salary discrimination in the USA professional baseball sports. Therefore, a RE model is selected for regression analysis in this study. In Eq (1), the subscript i denotes the i-th player; the subscript t denotes the t-th competition season; $u_i$ denotes the fixed effect; to identify the unobservable individual difference in the sample, $Nat_i$ is a time-invariant dummy variable, which denotes the nationality of the i-th player (if foreign MLB players suffer salary discrimination, the coefficient $\beta$ is smaller than 0; otherwise, the coefficient $\beta$ is larger than 0). $CD_i$ denotes the player's CD, which is measured using two methods in this study. The first method is the calculation method specified by Hofstede [4], specifically, Hofstede's six-dimension composite cultural indicator between the country of birth and the US; the second method is based on the GPS data specified by Falk et al. [8]. The CD can be calculated according to Eqs (2) and (3) wherein $P_{it}$ denotes the player's personal performance; $C_{it}$ denotes the player's personal characteristics; $v_{it}$ denotes the RE, which is an unobservable time variable in the sample; and $\varepsilon_{it}$ is the error term.

Eq (2) is the first method for measuring the CD, and is based on the study of Hofstede [4], involving six cultural dimensions. PDI denotes the degree to which organizational or social members accept the unequal distribution of power, namely, the degree to which people in a cultural context accept authority or privilege; IDV/COL denotes whether a society overall cares more about individual or collective interests; UAI denotes the degree to which a society can overall tolerate uncertainty; MAS/FEM denotes the degree to which masculinity is dominant in a cultural context; LTO/STO denotes the trade-off between long-term and short-term interests in a community; and IVR denotes the degree to which social members intend to restrain their desires.

$$CD_i = \sum_{k=1}^{6} \left\{ \left( I_{kj} - I_{kt} \right)^2 / V_k \right\} / 6. \tag{2}$$

where, $CD_i$ denotes the Hofstede CD of the i-th player between his/her country of birth (j) and the USA (t); $I_{kj}$ denotes the k-th cultural dimension indicator of a player in his/her country of birth (j); $I_{kt}$ denotes the k-th cultural dimension indicator of the USA (t); and $V_k$ denotes the variance of the k-th cultural dimension indicator[3].

$$CD_i = |Z_{kj} - Z_{kt}|. \tag{3}$$

The second equation for measuring the CD (i.e., Eq (3)) is based on the GPS data [8]. $CD_i$ denotes the GPS-based CD of the i-th player between his/her country of birth (j) and the USA (t); $z_{kj}$ denotes the preference of the country of birth (j) in the k-th GPS indicator; and $z_{kt}$ denotes the preference of the USA (t) in the k-th GPS indicator.

Professional sport is selected as the field of study because large amounts of player data are publicly available. Specifically, player data on salaries, personal characteristics, and performance can be cited from Baseball-Reference.com. The CD, which is based on Hofstede's six-dimension composite cultural indicator between players' country of birth and the US, can be cited from hofstede-insights.com, and can be calculated using an equation of Hofstede [4]. The representative data on altruism (ALT) and risk preference (RIS) can be cited from the GPS website (briq-institute.org/global-preferences/home). The GPS was conducted concurrently with the Gallup World Poll 2012, and aimed to measure preferences from the nationally representative samples of 76 countries, determine country-level average preference values, and examine the change in country-specific preferences. Each country-specific sample comprised 1,000 respondents, who were selected through probability sampling. The GPS questions

pertained to the preferences of more than 80,000 respondents worldwide for the six dimensions including power distance, risk preference, positive reciprocity, negative feedback, altruism, and trust. In this study, the preferences for altruism and risk preference were estimated.

The dependent variable is the logarithm of players' average annual salary (unit: $ 1 million), and the two major independent variables include nationality (NAT) and cultural distance (CD). Specifically, NAT is a dummy variable, which is distinguished by a player's place of birth. CD, including Hofstede's CD and GPS-based CD, is used to examine the impact on salaries from the preference difference between foreign players and American players.

## 3.1 Descriptive statistics

In this study, we selected a sample of 30 MLB teams, and collected 5,956 observations of 3,267 pitchers and 2,679 fielders during the sample period (i.e., four competition seasons from 2017 to 2020). The average salary of all players over the four seasons was $ 3.042 million, with Stephen Strasburg as the highest-salaried player. Stephen Strasburg, a US-born right-handed pitcher, earned as high as $ 38.333 million in 2019, 12.60 times the average. The MLB has established a base salary, with the lowest salary of $ 535,000 in 2017, which was earned by 9.39% of all MLB players during the sample period.

In the four seasons, foreign players accounted for 25%, and were mostly from North or South America. During the sample period, Panama had the largest CD from the U.S (3.89) whereas Australia had the smallest CD from the USA (0.02). In terms of uncertainty avoidance, Japan had the largest difference from the USA (46); in terms of collectivism, Panama had the largest difference from the USA (80); in terms of risk preference, South Africa had the largest difference from the USA (0.854); in terms of altruism, Mexico had the largest difference from the USA (1.22). The average age of players was 27.92 and the average tenure of players was 4.95 years. Table 1 offers the data on the performance of pitchers and fielders.

The sample is divided into American players and foreign players (in Table 2). The rightmost column lists the ratios of related variable values between foreign players and American players. The average salary of foreign players is $ 3.21 million, which is 1.08 times that of American players. The Win value of foreign players is 1.01 times that of American players. Among the seven pitcher performance indicators, there are five indicators in which foreign players outperform American players; among the six fielder performance indicators, there are four indicators in which foreign players outperform American players. Overall, most foreign fielders or pitchers outperform American players. Table 2 lists other data on the profile and performance of MLB players.

## 4 Analysis of empirical results

Players' performance is measured in terms of different indicators, depending on their position in baseball games. Therefore, the performance indicators of pitchers and fielders are analyzed respectively. Multiple performance indicators are highly correlated. To screen different combinations of player performance variables, a correlation coefficient of 0.7 is used as a criterion for player performance to prevent the estimation deviation due to a high degree of multicollinearity.

Table 3 lists the regression analysis results of pitchers by RE models. In Models 1 and 3, NAT is used as the major variable; in Models 2 and 4, the six-dimension composite cultural indicator is used as the major variable. In Models 3 and 4, the associated baseball league, team, and year are controlled to conduct a robustness test on major variables. In addition, the performance indicators are different between the models. The proxy variables for the performance of pitchers include Win, SHO, LOSE, HRA, CG, INNING, and BBP.

**Table 1. Descriptive statistics (N = 5,956).**

| Variable | Def | Description | Mean | Std. Dev. | Min | Max |
|---|---|---|---|---|---|---|
| salarymillion | Salary (unit: $ 1 million) | Annual salary of the player (unit: $ 1 million) | 3.04 | 5.23 | 0.54 | 38.33 |
| NAT | Nationality | Dummy variable (0: American player; 1: foreign player) | 0.25 | 0.44 | 0.00 | 1.00 |
| CD | Hofstede's six-dimension composite cultural indicator | Six-dimension composite cultural indicator between the country of birth and USA | 0.53 | 1.01 | 0.00 | 3.89 |
| UAI | Absolute value of uncertainty avoidance index | Based on Hofstede's CD; gap in uncertainty avoidance between players | 4.18 | 10.50 | 0.00 | 46.00 |
| COL | Absolute value of collectivism | Based on Hofstede's CD; gap in collectivism between players | 17.02 | 29.50 | 0.00 | 80.00 |
| RIS | Absolute value of risk preference | Based on GPS data; gap in risk preference between players | 0.03 | 0.08 | 0.00 | 0.85 |
| ALT | Absolute of altruism | Based on GPS data; gap in altruism between players | 0.06 | 0.19 | 0.00 | 1.22 |
| Player characteristics | | | | | | |
| AGE | Age | Age of the player | 27.92 | 3.67 | 19.00 | 45.00 |
| SQAGE | Square of age | Square of age of the player | 793.14 | 213.81 | 361.00 | 2025.00 |
| Tenure | Tenure | Tenure of the player | 4.95 | 3.65 | 1.00 | 21.00 |
| SQTenure | Square of tenure | Square of tenure of the player | 37.74 | 52.55 | 1.00 | 441.00 |
| BMI | Body mass index | BMI of the player | 27.44 | 2.34 | 19.58 | 40.33 |
| SQBMI | Square of body mass index | Square of BMI of the player | 758.47 | 131.04 | 383.51 | 1626.42 |
| Pitcher performance | | | | | | |
| Win | Number of wins | Number of wins attained by the player in the current season | 2.55 | 3.92 | 0.00 | 104.00 |
| LOSE | Number of losses | Number of losses by the player in the current season | 2.63 | 5.82 | 0.00 | 227.00 |
| CG | Number of completed games | Number of times that the player takes the field to pitch in the current season | 0.06 | 0.34 | 0.00 | 8.00 |
| SHO | Number of shutouts | Number of times that the player shuts the opponent out in the current season | 0.06 | 0.99 | 0.00 | 38.00 |
| INNING | Number of innings | Number of pitch innings attended by the player in the current season | 44.28 | 47.40 | 0.00 | 223.00 |
| HRA | Number of home runs | Number of home runs by the planer in the current season | 6.34 | 7.04 | 0.00 | 41.00 |
| BBP | Number of bases on balls | Number of bases on balls achieved by the player in the current season | 16.31 | 15.96 | 0.00 | 95.00 |
| Fielder performance | | | | | | |
| AB | At bat | Number of times that the player completes the hits other than successful sacrifice hits, bases on balls, hits by pitch, or sacrifice fly in the current season | 202.25 | 182.43 | 0.00 | 681.00 |
| DB | Number of two-base bits | Number of two-base bits achieved by the player in the current season while the fielder makes no fault | 10.35 | 10.79 | 0.00 | 56.00 |
| SB | Number of stolen bases | Number of times that the runner successfully reaches the next base in the pitching process | 3.05 | 5.76 | 0.00 | 60.00 |
| CS | Number of caught stealings | Number of times that the base stealing is shut out by the fielder in the current season | 1.12 | 1.87 | 0.00 | 16.00 |
| FB | Number of bases on balls | Number of bases on balls in the current season | 1.07 | 2.28 | 0.00 | 25.00 |
| RS | Score | Number of times that the player returns to the home base safely in the current season | 28.06 | 28.38 | 0.00 | 137.00 |

Table 4 lists the regression analysis results of fielders by RE models. In Models 1 and 3, NAT is used as the major variable; in Models 2 and 4, the six-dimension composite cultural indicator is used as the major variable. In Models 3 and 4, the associated baseball league, team and year are controlled. In addition, the performance indicators are different between the models. The proxy variables for the performance of fielders include AB, DB, SB, CS, FB, and RS. The objective is to check whether different performance indicators are added to produce different results and affect the most important variables (i.e., NAT and CD) of this study.

**Table 2. Descriptive statistics of American players and foreign players.**

| Variable | Def | American players | | Foreign players | | Foreign players / American players |
|---|---|---|---|---|---|---|
| | | Mean | Std. Dev. | Mean | Std. Dev. | Ratio |
| salarymillion | Salary (unit: $ 1 million) | 2.98 | 5.27 | 3.21 | 5.13 | 1.08 |
| CD | Hofstede's six-dimension composite cultural indicator | - | - | 2.11 | 0.93 | / |
| UAI | Absolute value of uncertainty avoidance index | - | - | 16.96 | 15.75 | / |
| COL | Absolute value of collectivism | - | - | 66.34 | 12.89 | / |
| RIS | Absolute value of risk preference | - | - | 0.19 | 0.13 | / |
| ALT | Absolute of altruism | - | - | 0.44 | 0.30 | / |
| Player characteristics | | | | | | |
| AGE | Age | 28.08 | 3.48 | 27.45 | 4.16 | 0.98 |
| SQAGE | Square of age | 800.77 | 202.19 | 770.82 | 243.35 | 0.96 |
| Tenure | Tenure | 4.91 | 3.53 | 5.04 | 3.96 | 1.03 |
| SQTenure | Square of tenure | 36.59 | 48.43 | 41.10 | 62.98 | 1.12 |
| BMI | Body mass index | 27.17 | 2.09 | 28.22 | 2.80 | 1.04 |
| SQBMI | Square of body mass index | 742.76 | 114.77 | 804.44 | 161.39 | 1.08 |
| Pitcher performance | | | | | | |
| Win | Number of wins | 2.54 | 3.49 | 2.58 | 5.13 | 1.01 |
| LOSE | Number of losses | 2.51 | 3.11 | 3.05 | 10.88 | 1.22 |
| CG | Number of completed games | 0.05 | 0.28 | 0.07 | 0.49 | 1.28 |
| SHO | Number of shutouts | 0.04 | 0.77 | 0.10 | 1.51 | 2.47 |
| INNING | Number of innings | 44.61 | 47.72 | 43.12 | 46.27 | 0.97 |
| HRA | Number of home runs | 6.38 | 7.00 | 6.21 | 7.17 | 0.97 |
| BBP | Number of bases on balls | 16.32 | 15.92 | 16.28 | 16.10 | 1.00 |
| Fielder performance | | | | | | |
| AB | At bat | 193.36 | 178.75 | 223.56 | 189.39 | 1.16 |
| DB | Number of two-base bits | 9.96 | 10.69 | 11.28 | 10.96 | 1.13 |
| SB | Number of stolen bases | 2.94 | 5.78 | 3.30 | 5.70 | 1.12 |
| CS | Number of caught stealings | 1.06 | 1.77 | 1.27 | 2.07 | 1.20 |
| FB | Number of bases on balls | 1.08 | 2.41 | 1.04 | 1.96 | 0.97 |
| RS | Score | 27.24 | 28.37 | 30.03 | 28.31 | 1.10 |

## 4.1 Regression analysis of pitchers by RE models

The NAT coefficient is significantly positively correlated with pitcher salaries in both Model 1 and Model 3, with the coefficient values ranging from 0.18 to 0.19 (in Table 3). Specifically, while other conditions remain unchanged, foreign pitchers are paid 0.18% to 0.19% higher than American pitchers, proving the existence of salary premium. The CD coefficient is also significantly positively correlated with pitcher salaries in both Model 2 and Model 4. Specifically, while other conditions remain unchanged, a 1-unit increase in CD brings about an approximately 0.070% to 0.071% increase in pitcher salaries. The Age coefficient is significantly positively correlated with pitcher salaries, and SQAGE is significantly negatively correlated with pitcher salaries; specifically, a 1-year increase in age brings about a 0.27% to 0.29% increase in pitcher salaries at a diminishing rate. The Tenure coefficient is also significantly positively correlated with pitcher salaries, and SQTenure is significantly negatively correlated with pitcher salaries; specifically, a 1-year increase in tenure brings about an approximately 0.28% increase in pitcher salaries, except that this phenomenon is weakening with the increase in tenure.

**Table 3. Regression analysis of pitchers by RE models.**

| Dependentvariables: logsal | | | | |
|---|---|---|---|---|
| **VARIABLES** | **Model 1** | **Model 2** | **Model 3** | **Model 4** |
| Player characteristics | | | | |
| NAT | 0.19*** | | 0.18*** | |
| | (0.049) | | (0.050) | |
| CD | | 0.071*** | | 0.070*** |
| | | (0.017) | | (0.017) |
| AGE | 0.27*** | | 0.29*** | |
| | (0.053) | | (0.054) | |
| SQAGE | -0.0019** | | -0.0023** | |
| | (0.00091) | | (0.00093) | |
| BMI | 0.060 | 0.055 | 0.063 | 0.055 |
| | (0.10) | (0.085) | (0.11) | (0.082) |
| SQBMI | -0.0011 | -0.0013 | -0.0012 | -0.0013 |
| | (0.0019) | (0.0015) | (0.0019) | (0.0015) |
| Tenure | | 0.28*** | | 0.28*** |
| | | (0.012) | | (0.012) |
| SQTenure | | -0.0054*** | | -0.0055*** |
| | | (0.00091) | | (0.00090) |
| Performance | | | | |
| Win | | 0.018*** | 0.020*** | |
| | | (0.0040) | (0.0038) | |
| SHO | -0.012 | | -0.016 | 0.0018 |
| | (0.014) | | (0.013) | (0.012) |
| Lose | 0.0031 | 0.00040 | | |
| | (0.0031) | (0.0028) | | |
| HRA | 0.014*** | | | |
| | (0.0020) | | | |
| CG | 0.097** | 0.061 | 0.093** | 0.071* |
| | (0.039) | (0.037) | (0.039) | (0.037) |
| INNING | | | | 0.0025*** |
| | | | | (0.00028) |
| League | | | Yes | Yes |
| Team | | | Yes | Yes |
| Year | | | Yes | Yes |
| Constant | 6.97*** | 12.2*** | 6.61*** | 12.1*** |
| | (1.52) | (1.17) | (1.54) | (1.13) |
| Observations | 3,229 | 3,192 | 3,229 | 3,192 |
| Number of player_id | 1,230 | 1,216 | 1,230 | 1,216 |
| R-squared | 0.422 | 0.636 | 0.413 | 0.667 |

Notes: 1. Standard errors in parentheses

*** $p < 0.01$,

** $p < 0.05$,

* $p < 0.1$.

2. Most research in this area uses both cluster correction and a correction for heteroscedasticity. Therefore, our results using a team level cluster correction and a correction for heteroscedasticity are employed and the results support the previous findings.

**Table 4. Regression analysis of fielders by RE models.**

| Dependentvariables: logsal | | | | |
|---|---|---|---|---|
| VARIABLES | Model 1 | Model 2 | Model 3 | Model 4 |
| Player characteristics | | | | |
| NAT | 0.078* | | 0.087* | |
| | (0.046) | | (0.046) | |
| CD | | 0.057** | | 0.053** |
| | | (0.022) | | (0.022) |
| BMI | 0.23* | 0.24 | 0.20 | 0.21 |
| | (0.13) | (0.14) | (0.12) | (0.14) |
| SQBMI | -0.0040* | -0.0041 | -0.0034 | -0.0036 |
| | (0.0022) | (0.0025) | (0.0022) | (0.0025) |
| AGE | | 0.46*** | | 0.42*** |
| | | (0.063) | | (0.063) |
| SQAGE | | -0.0050*** | | -0.0044*** |
| | | (0.0011) | | (0.0011) |
| Tenure | 0.32*** | | 0.31*** | |
| | (0.015) | | (0.016) | |
| SQTenure | -0.0081*** | | -0.0072*** | |
| | (0.0010) | | (0.0011) | |
| Performance | | | | |
| FB | 0.0093 | 0.012 | 0.011 | 0.015* |
| | (0.0075) | (0.0083) | (0.0075) | (0.0083) |
| AB | 0.00098*** | 0.0013*** | 0.0012*** | 0.0021*** |
| | (0.00011) | (0.00035) | (0.00012) | (0.00035) |
| DB | | -0.0049 | | -0.0047 |
| | | (0.0042) | | (0.0042) |
| CS | | -0.033** | | -0.030** |
| | | (0.013) | | (0.013) |
| RS | | 0.0026 | | -0.00066 |
| | | (0.0022) | | (0.0022) |
| League | | | YES | YES |
| Team | | | YES | YES |
| Year | | | YES | YES |
| Constant | 9.30*** | 1.55 | 9.66*** | 2.45 |
| | (1.78) | (2.20) | (1.77) | (2.17) |
| Observations | 2,639 | 2,531 | 2,639 | 2,531 |
| Number of player_id | 935 | 901 | 935 | 901 |
| R-squared | 0.626 | 0.440 | 0.652 | 0.546 |

Notes: 1. Standard errors in parentheses

*** $p < 0.01$,

** $p < 0.05$,

* $p < 0.1$. 2. Most research in this area uses both cluster correction and a correction for heteroscedasticity. Therefore, our results using a team level cluster correction and a correction for heteroscedasticity are employed and the results support the previous findings.

The Win coefficient is significantly positively correlated with pitcher salaries in Models 1 to 4; specifically, an increase of 1 win brings about a 0.018% to 0.020% increase in pitcher salaries. The CG coefficient is significantly positively correlated with pitcher salaries in Models 1 to 4; specifically, an increase of 1 completed game brings about a 0.071% to 0.097% increase in

pitcher salaries. The INNING coefficient is significantly positively correlated with pitcher salaries in Models 1 to 4; specifically, an increase of 10 innings brings about an approximately 0.025% increase in pitcher salaries.

Table 5 lists the regression analysis results of pitchers by RE models. RIS is significantly positively correlated with pitcher salaries in Models 1 to 5, with the coefficient values ranging from 1.05 to 1.88. Specifically, while other conditions remain unchanged, the risk preference difference between foreign and American pitchers brings about a 1.05% to 1.88% increase in the salaries of foreign pitchers. UAI is also significantly positively correlated with pitcher salaries in Models 6 to 10.

COL is significantly negatively correlated with pitcher salaries, with the coefficient values ranging from -0.0017 to -0.0018. Specifically, while other conditions remain unchanged, the collectivism difference between foreign and American pitchers brings about a 0.0017% to 0.0018% decrease in the salaries of foreign pitchers. ALT is significantly negatively correlated with pitcher salaries, with the coefficient values ranging from -0.42 to -0.46. Specifically, while other conditions remain unchanged, the altruism difference between foreign and American pitchers brings about a 0.42% to 0.46% decrease in the salaries of foreign pitchers.

## 4.2 Regression analysis of fielders by RE models

The NAT coefficient is significantly positively correlated with fielder salaries in both Model 1 and Model 3, with the coefficient values ranging from 0.078 to 0.087 (in Table 4). Specifically, while other conditions remain unchanged, foreign fielders are paid 0.078% to 0.087% higher than American fielders, proving the existence of salary premium. The CD coefficient is significantly positively correlated with fielder salaries in both Model 2 and Model 4. Specifically, while other conditions remain unchanged, a 1-unit increase in CD brings about a 0.053% to 0.057% increase in fielder salaries. BMI is significantly positively correlated with fielder salaries, and SQBMI is significantly negatively correlated with fielder salaries; specifically, a 1-unit increase in BMI brings 0.23% increase in fielder salaries, except that this phenomenon is weakening with the increase in BMI. AGE is significantly positively correlated with fielder salaries, and SQAGE is significantly negatively correlated with fielder salaries; specifically, a 1-year increase in age brings about a 0.42% to 0.46% increase in fielder salaries at a diminishing rate. Tenure is also significantly positively correlated with fielder salaries, and SQTenure is significantly negatively correlated with fielders' salaries; specifically, a 1-year increase in tenure brings about an approximately 0.31% to 0.32% increase in fielder salaries at a diminishing rate.

Among the fielder performance variables, AB is significantly positively correlated with fielder salaries; specifically, an increase of 10 at-bat times brings about a 0.00099% to 0.0021% increase in fielder salaries. CS is significantly negatively correlated with fielder salaries; specifically, a 1-unit increase in CS brings about a 0.030% to 0.033% decrease in fielder salaries.

Table 6 lists the regression analysis results of fielders by RE models. In Models 1 to 5, ALT is significantly positively correlated with fielder salaries, with the coefficient values ranging from 0.53 to 0.63. Specifically, while other conditions remain unchanged, the altruism difference between foreign and American fielders brings about a 0.53% to 0.63% increase in the salaries of foreign fielders. In Models 6 to 10, COL is significantly positively correlated with fielder salaries, with the coefficient values ranging from 0.0028 to 0.0032. Specifically, while other conditions remain unchanged, the collectivism difference between foreign and American fielders brings about a 0.0028% to 0.0032% increase in the salaries of foreign fielders.

Many previous studies have demonstrated the race-related or nationality-related salary discrimination in professional sports [1–3, 13, 16, 22], and their estimation results are compared with each other in Table A in S1 Appendix. Regarding the CD-related salary estimation in

**Table 5. Regression analysis of pitchers' risk preference (uncertainty avoidance) and altruism (collectivism) by RE models.**

| | Dependent variables: logsal | | | | | | | | | |
|---|---|---|---|---|---|---|---|---|---|---|
| **VARIABLES** | **Model 1** | **Model 2** | **Model 3** | **Model 4** | **Model 5** | **Model 6** | **Model 7** | **Model 8** | **Model 9** | **Model 10** |
| **Player characteristics** | | | | | | | | | | |
| RIS/UAI | 1.88*** | 1.05** | 1.84*** | 1.06*** | 1.88*** | 0.012*** | 0.0099*** | 0.011*** | 0.0095*** | 0.011*** |
| | (0.33) | (0.41) | (0.32) | (0.41) | (0.32) | (0.0021) | (0.0027) | (0.0021) | (0.0027) | (0.0021) |
| ALT /COL | -0.42*** | -0.024 | -0.44*** | -0.063 | -0.46*** | -0.0018** | 0.00016 | -0.0017** | 0.00016 | -0.0017** |
| | (0.13) | (0.17) | (0.13) | (0.17) | (0.13) | (0.00080) | (0.0010) | (0.00080) | (0.0010) | (0.00079) |
| Control vble. [a] | YES | YES | YES | YES | YES | YES | YES | YES | YES | YES |
| League | | | YES | YES | YES | | | YES | YES | YES |
| Team | | | YES | YES | YES | | | YES | YES | YES |
| Year | | | YES | YES | YES | | | YES | YES | YES |
| Constant | 12.4*** | 8.58*** | 12.5*** | 8.13*** | 12.2*** | 12.9*** | 7.29*** | 13.1*** | 7.03*** | 12.9*** |
| | (1.54) | (2.03) | (1.53) | (2.02) | (1.52) | (1.15) | (1.51) | (1.15) | (1.50) | (1.14) |
| Observations | 2,779 | 2,779 | 2,780 | 2,779 | 2,780 | 3,185 | 3,185 | 3,186 | 3,185 | 3,186 |
| Number of player_id | 1,051 | 1,051 | 1,051 | 1,051 | 1,051 | 1,213 | 1,213 | 1,213 | 1,213 | 1,213 |
| R-squared | 0.6034 | 0.4038 | 0.6124 | 0.4247 | 0.6164 | 0.6585 | 0.4265 | 0.6639 | 0.4477 | 0.6697 |

Notes: 1. Standard errors in parentheses

*** $p < 0.01$,

** $p < 0.05$,

* $p < 0.1$. 2. Most research in this area uses both cluster correction and a correction for heteroscedasticity. Therefore, our results using a team level cluster correction and a correction for heteroscedasticity are employed and the results support the previous findings. 3.

[a] Other control variables include AGE, SQAGE, Tenure, SQTenure, BMI, SQBMI, CG, SHO, WIN, LOSE, HRA, INNING, BBP, and the result will be provided by author if needed.

professional sports, Jane analyzed the salaries of Japan's professional baseball players, finding that the salary premium of players from Holland with the largest CD (6.44) is 10.45% higher than that of players from Italy with the smallest CD (0.94) [5].

Moreover, other cross-cultural studies cover topics such as salary gap in multinational companies [24, 25], impact of cultural background on financial and non-financial rewards [26], and professional players [5]. Many studies focused on salary gap caused by CD between company employees, but few studies have discussed the salary gap from the perspective of cross-cultural background in sports leagues in detail. In this study, we introduced related cultural indicators of players to examine salary gap caused by cultural differences between players and further estimate the salary premium of MLB players with different nationalities. As described in Table 7, players with a salary premium of not more than 19.73% are sequentially from South Korea, Panama, Japan, Saudi Arabia, and Venezuela. Players with the lowest salary premium (0.11%) are from Australia, followed by Canada, United Kingdom, South Africa, and Germany. The premium on salaries of Australian players who have the smallest CD (0.02) from the USA is 19.61% (= (3.72–0.02)*5.3%) higher than that of players from South Korea with the largest CD (3.72) from the USA.

The above analysis offsets the deficiency in studies that examined the salary discrimination in professional sports from a culture perspective. However, this study did not analyze the accurate salary gap between players with different nationalities, or estimate the salary gap using the QR method. In addition, this study merely introduced two cultural dimensions (including altruism and risk preference) of the six cultural dimensions in the GPS. Other cultural dimensions (e.g., power distance and trust) may help further explain the impact of cultural

**Table 6. Regression analysis of fielders' risk preference (uncertainty avoidance) and altruism (collectivism) by RE models.**

| VARIABLES | Dependent variables: logsal | | | | | | | | | |
|---|---|---|---|---|---|---|---|---|---|---|
| | Model 1 | Model 2 | Model 3 | Model 4 | Model 5 | Model 6 | Model 7 | Model 8 | Model 9 | Model 10 |
| **Player characteristics** | | | | | | | | | | |
| RIS /UAI | -0.71 | -0.66 | -0.51 | -0.55 | -0.54 | -0.0033 | -0.0029 | -0.0025 | -0.0027 | -0.0028 |
| | (0.58) | (0.58) | (0.58) | (0.58) | (0.57) | (0.0033) | (0.0033) | (0.0033) | (0.0033) | (0.0032) |
| ALT /COL | 0.62** | 0.63** | 0.56** | 0.54** | 0.53** | 0.0030** | 0.0032*** | 0.0031*** | 0.0028** | 0.0028** |
| | (0.27) | (0.27) | (0.27) | (0.27) | (0.27) | (0.0012) | (0.0012) | (0.0012) | (0.0012) | (0.0012) |
| Control vble. [a] | YES | YES | YES | YES | YES | YES | YES | YES | YES | YES |
| League | | | YES | YES | YES | | | YES | YES | YES |
| Team | | | YES | YES | YES | | | YES | YES | YES |
| Year | | | YES | YES | YES | | | YES | YES | YES |
| Constant | 0.84 | 0.76 | 1.49 | 1.19 | 1.73 | 1.50 | 1.41 | 2.16 | 2.03 | 2.44 |
| | (2.47) | (2.47) | (2.47) | (2.47) | (2.44) | (2.20) | (2.21) | (2.19) | (2.19) | (2.17) |
| Observations | 2,144 | 2,144 | 2,144 | 2,144 | 2,144 | 2,529 | 2,529 | 2,529 | 2,529 | 2,529 |
| Number of player_id | 764 | 764 | 764 | 764 | 764 | 900 | 900 | 900 | 900 | 900 |
| R-squared | 0.4186 | 0.4217 | 0.4591 | 0.4570 | 0.4676 | 0.4784 | 0.4800 | 0.5332 | 0.5368 | 0.5472 |

Notes: 1. Standard errors in parentheses

*** p<0.01,

** p<0.05,

* p<0.1. 2. Most research in this area uses both cluster correction and a correction for heteroscedasticity. Therefore, our results using a team level cluster correction and a correction for heteroscedasticity are employed and the results support the previous findings. 3.

[a] Other control variables include AGE, SQAGE, BMI, SQBMI, SB, FB, AB, DB, CS, RS, and the result will be provided by author if needed.

differences. Subsequent studies may verify the above conclusions and conduct an in-depth analysis considering the mutual effect of diverse complex factors.

## 5 Conclusions

This study discusses the salary discrimination among the MLB players from the perspective of nationality and CD. Based on the long-term tracking data (panel data) of a sample of 3,267 pitchers and 2,679 fielders from 30 MLB teams over four seasons from 2017 to 2020, we conducted regression analyses on 5,965 observations using RE models. First, the analysis results showed that NAT was significantly positively correlated with player salaries; this finding is consistent with the findings of previous studies. Then, we discussed the salary discrimination from a cross-cultural perspective, and calculated the CD of players between their countries of birth and USA, finding that CD was positively correlated with player salaries. Moreover, we found that both foreign pitchers and foreign fielders enjoyed a salary premium over American players.

In addition to the six-dimension composite cultural indicator (i.e., CD), we conducted separate regression analysis on two similar dimensions, including Hofstede's collectivism (or uncertainty avoidance) and altruism (or risk preference) in the GPS, to determine the impact of individual preference difference in different countries on salaries. The major findings are summarized as three points. First, the salaries of baseball players vary with their roles and risk attitudes. The difference in risk preference and uncertainty avoidance is significantly positively correlated with the salaries of pitchers; specifically, the larger the risk preference difference between foreign and American pitchers, the higher the salary premium of foreign pitchers. Hence, it is advisable to employ foreign players who share similar risk preferences with

**Table 7. Estimation of salary premium of MLB players with different nationalities.**

| Rank | Country | Salary Premium | Cultural Distance |
|------|---------|----------------|-------------------|
| 1 | USA | - | - |
| 2 | South Korea | 19.73% | 3.72 |
| 3 | Panama | 19.38% | 3.66 |
| 4 | Japan | 15.84% | 2.99 |
| 5 | Saudi Arabia | 15.58% | 2.94 |
| 6 | Venezuela | 15.29% | 2.88 |
| 7 | Lithuania | 14.58% | 2.75 |
| 8 | Taiwan | 13.79% | 2.60 |
| 9 | Peru | 13.72% | 2.59 |
| 10 | Mexico | 12.90% | 2.43 |
| 11 | Hong Kong | 12.75% | 2.41 |
| 12 | Columbia | 11.89% | 2.24 |
| 13 | Honduras | 10.93% | 2.06 |
| 14 | Nicaragua | 8.79% | 1.66 |
| 15 | Netherlands | 8.59% | 1.62 |
| 16 | Brazil | 8.32% | 1.57 |
| 17 | Puerto Rico | 8.10% | 1.53 |
| 18 | Dominican Republic | 6.13% | 1.16 |
| 19 | Germany | 5.93% | 1.12 |
| 20 | South Africa | 1.07% | 0.20 |
| 21 | United Kingdom | 1.00% | 0.19 |
| 22 | Canada | 0.53% | 0.10 |
| 23 | Australia | 0.11% | 0.02 |

American players (e.g., players from Canada, Dominican Republic, and Australia), to avoid paying an excessive salary premium. Second, collectivism and altruism are significantly negatively correlated with the salaries of pitchers; hence, it is advisable to employ players from countries that are significantly different from the USA in terms of collectivism. Third, collectivism and altruism are significantly positively correlated with the salaries of fielders; hence, fielders should cooperate closely in baseball games to achieve better performance. Taking the offensive tactics sacrifice hit as an example, the player uses a surrender of one out in exchange for base advance. When there is a runner on the base but there is no out or one out, the fielder uses a sacrifice hit to advance the base runner to the scoring position at the cost of his/her own out, thus escorting his teammates to the base or scoring a run through an altruistic team spirit.

The above results show that the salary premium given to foreign players may be due to the limited information on international players available to employers, or due to blind faith in foreign players; this is known as salary discrimination due to information asymmetry. It is not advisable to pay an excessive salary premium to foreign players. While other conditions remain unchanged, players from South Korea and Panama enjoy a nearly 20% salary premium, whereas the salary premium of players from Australia, Canada, United Kingdom, South Africa, and Germany is the lowest. The salary premium for players from Dominican Republic (accounting for the largest proportion of foreign MLB players) is 6.13%. The objective is to help the MLB and baseball teams review their foreign player recruitment policy, avoid reducing the salaries of American players with the same performance as foreign players, and allocate funds among players properly. The findings of this study can provide a reference for the recruitment policy in the professional baseball industry in different countries.

## Supporting information

**S1 Appendix.**
(DOCX)

**S1 Text.**
(DOCX)

**S1 File. Description: The raw data in this research is collected from various sources.** Specifically, player data on salaries, personal characteristics, and performance can be cited from *Baseball-Reference.com*. The CD, which is based on Hofstede's six-dimension composite cultural indicator between players' country of birth and the US, can be cited from *hofstede-insights.com*.
(ZIP)

**S2 File. Description: The raw data in this research is collected from various sources.** Specifically, player data on salaries, personal characteristics, and performance can be cited from *Baseball-Reference.com*. The CD, which is based on Hofstede's six-dimension composite cultural indicator between players' country of birth and the US, can be cited from *hofstede-insights.com*.
(ZIP)

## Author Contributions

**Writing – original draft:** Wen-Jhan Jane, Yi-Jie Yu, Jye-Shyan Wang.

**Writing – review & editing:** Wen-Jhan Jane, Jye-Shyan Wang.

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
