## [Decision Letter · Decision Letter 0]

31 Oct 2022

PONE-D-22-24132The Impacts of National Culture, Altruism, and Risk Preference on Salaries: The Case of the Major League BaseballPLOS ONE

Dear Dr. Wang,

Thank you for submitting your manuscript to PLOS ONE. After careful consideration, we feel that it has merit but does not fully meet PLOS ONE’s publication criteria as it currently stands. Therefore, we invite you to submit a revised version of the manuscript that addresses the points raised during the review process.

We look forward to receiving your revised manuscript.

Kind regards,

Muhammad Fareed, Ph.D

Academic Editor

PLOS ONE

Journal Requirements:

2. The Dominican Republic and Dominica are two separate nations: You make several references to Dominica in your manuscript, but it is the Dominican Republic (whose flag you present in Fig 1) that produces many baseball players. Please check this, and change "Dominica" to "the Dominican Republic" if appropriate.

4. Please ensure that you refer to Figure 1 in your text as, if accepted, production will need this reference to link the reader to the figure.

Additional Editor Comments:

The authors should take note of the following:

-> Update the literature review with more recent articles.

-> Carry out a comprehensive proofreading of the whole manuscript.

-> Cross-check the consistency of the citation with the references.

-> Impacts word may be replaced with impact

-> The literature review may be structured better with sub-topics on different variables of study

-> The authors have not generated hypotheses, it is important to do so

-> Add a research model for clarity

-> The tables may be made more concise and presented better, some may be removed and added in the appendix section

-> A section on discussion and implications should be added before conclusion

-> Also mention scope for future research

-> It is a very lengthy paper and may not keep the reader interested, the paper hence needs to be trimmed down

Reviewers' comments:

Reviewer's Responses to Questions

**Comments to the Author**

1. Is the manuscript technically sound, and do the data support the conclusions?

Reviewer #1: Yes

Reviewer #2: Yes

2. Has the statistical analysis been performed appropriately and rigorously? 

Reviewer #1: Yes

Reviewer #2: Yes

3. Have the authors made all data underlying the findings in their manuscript fully available?

Reviewer #1: Yes

Reviewer #2: No

4. Is the manuscript presented in an intelligible fashion and written in standard English?

Reviewer #1: Yes

Reviewer #2: Yes

5. Review Comments to the Author

Reviewer #1: Impacts word may be replaced with impact

The literature review may be structured better with sub-topics on different variables of study

The authors have not generated hypotheses, it is important to do so

Add a research model for clarity

The tables may be made more concise and presented better, some may be removed and added in the appendix section

A section on discussion and implications should be added before conclusion

Also mention scope for future research

It is a very lengthy paper and may not keep the reader interested, the paper hence needs to be trimmed down

Reviewer #2: The manuscript is publishable as it discusses an important research area. It also contains all the necessary components of a good research article. However, the authors should take note of the following:

o Update the literature review with more recent studies

o Carry out a comprehensive proofreading of the manuscript.

o Checking for the consistency of the citation with the references is also suggested.

Generally, to improve the quality of the manuscript, it is suggested for the authors to read a recent article on research methodology titled “Best practices in data collection and preparation: Recommendations for reviewers, editors, and authors” and integrate some of the recommendations of the authors of the article in their manuscript.

Thanks.

6. PLOS authors have the option to publish the peer review history of their article (what does this mean?). If published, this will include your full peer review and any attached files.

Reviewer #1: No

Reviewer #2: No

---

## [Author Response · Author response to Decision Letter 0]

27 Nov 2022

The manuscript is publishable as it discusses an important research area. It also contains all the necessary components of a good research article. However, the authors should take note of the following:

 Update the literature review with more recent studies 

Response: We include five recent studies in the text and update the references and literatures.

 Celik, O. B., & Ince-Yenilmez, M. Salary differences under the salary cap in Major League Soccer. International Journal of Sports Science & Coaching. 2017; 12(5) : 623-634. [15]

 Jung, H. Salary Discrimination in the Sports Labor Market: An Evidence from Major League Soccer. 2021; Doctoral dissertation, University of Northern Colorado. [18]

 Kerr, C. An industry test for ethnic discrimination in major league soccer. Applied Economics Letters. 2019; 26(16): 1358-1363. [19]

 Medcalfe, S., & Smith, R. The effect of foreign players on pay and performance in Major League Soccer. International Journal of Sport Finance. 2018; 13(4): 297-318. [20]

 Simmons, R. Professional labor markets in the Journal of Sports Economics. Journal of Sports Economics. 2022; 23(6): 728-748. [21]

The first paragraph of section 2.2 (Salary Discrimination in Other Professional Sports) on page 10 is rewritten as “In addition to the baseball sports, studies on salary discrimination have also been conducted in other sports [2,15,16,17,18,19, 20].” Regarding salary discrimination in football, Swift analyzed the data of 1,397 MLS players……“Simmons summaries the research findings of papers published in Journal of Sports Economics on the workings of professional sports labor markets. In sum, the evidence of hiring and exit discrimination in various leagues looks stronger than the evidence for salary discrimination. [21].”

 Carry out a comprehensive proofreading of the manuscript. 

Response: we had hired a professional editor to carry out a comprehensive proofreading of the manuscript. thank you for the valuable suggestioin.

 Checking for the consistency of the citation with the references is also suggested.

Response: we check the article over and over for the consistency of the citation with the references. Thank you for the reminder.

Generally, to improve the quality of the manuscript, it is suggested for the authors to read a recent article on research methodology titled “Best practices in data collection and preparation: Recommendations for reviewers, editors, and authors” and integrate some of the recommendations of the authors of the article in their manuscript.

Thanks.

Response:

Aguinis, Hill, & Bailey (2021) offer best-practice recommendations for journal reviewers, editors, and authors regarding data collection and preparation. According to their recommendations regarding data collection address, we check our data analysis again for the type of research design, control variables, sampling procedures, and missing data management. The raw data and codes in STATA are attached with the submission, thank you for the valuable suggestion. 

 

PONE-D-22-24132

The Impacts of National Culture, Altruism, and Risk Preference on Salaries: The Case of the Major League Baseball

PLOS ONE

Dear Dr. Wang,

Thank you for submitting your manuscript to PLOS ONE. After careful consideration, we feel that it has merit but does not fully meet PLOS ONE’s publication criteria as it currently stands. Therefore, we invite you to submit a revised version of the manuscript that addresses the points raised during the review process.

We look forward to receiving your revised manuscript.

Kind regards,

Muhammad Fareed, Ph.D

Academic Editor

PLOS ONE

Journal Requirements:

Response: We had edited the manuscript according to PLOS ONE's style requirements. 

Thank you for your reminder.

2. The Dominican Republic and Dominica are two separate nations: You make several references to Dominica in your manuscript, but it is the Dominican Republic (whose flag you present in Fig 1) that produces many baseball players. Please check this, and change "Dominica" to "the Dominican Republic" if appropriate.

Response: Thank you for the valuable suggestion. The term Dominica in the manuscript has been corrected. The related corrections are listed as follows.

 Abstract is rewritten as “Salary estimation data showed that the salary premium was nearly 20% for players from South Korea and Panama, the lowest (only 0.11%) for players from Australia, and only 6.13% for players from Dominican Republic (accounting for the largest proportion of foreign MLB players), indicating that the MLB’s foreign player recruitment policy is correct.”

 The first paragraph on page 4 is rewritten as “Among foreign players, as we presented in the Figure 1, 243 players from Dominican Republic (DO) account for 42%,…”

 The first paragraph on page 39 is rewritten as “Hence, it is advisable to employ foreign players who share similar risk preferences with American players (e.g., players from Canada, Dominican Republic, and Australia), to avoid paying an excessive salary premium. ”

 The first paragraph on page 40 is rewritten as “The salary premium for players from Dominican Republic (accounting for the largest proportion of foreign MLB players) is 6.13%.”

Response: Thank you for the valuable suggestion. We will submit the data (the raw data in the form of .dta and .do files in STATA) with the manuscript at this submission.

4. Please ensure that you refer to Figure 1 in your text as, if accepted, production will need this reference to link the reader to the figure.

Response: Thank you for the suggestion. The statement of Figure 1 has been inserted.

-The second paragraph on page 4 is rewritten as “Among foreign players, as presented in the Figure 1, 243 players from Dominican Republic (DO) account for 42%,…”

Additional Editor Comments:

The authors should take note of the following:

-> Update the literature review with more recent articles.

Response: We include five recent studies in the text and update the references and literatures.

 Celik, O. B., & Ince-Yenilmez, M. Salary differences under the salary cap in Major League Soccer. International Journal of Sports Science & Coaching. 2017; 12(5) : 623-634. [15]

 Jung, H. Salary Discrimination in the Sports Labor Market: An Evidence from Major League Soccer. 2021; Doctoral dissertation, University of Northern Colorado. [18]

 Kerr, C. An industry test for ethnic discrimination in major league soccer. Applied Economics Letters. 2019; 26(16): 1358-1363. [19]

 Medcalfe, S., & Smith, R. The effect of foreign players on pay and performance in Major League Soccer. International Journal of Sport Finance. 2018; 13(4): 297-318. [20]

 Simmons, R. Professional labor markets in the Journal of Sports Economics. Journal of Sports Economics. 2022; 23(6): 728-748. [21]

The first paragraph of section 2.2 (Salary Discrimination in Other Professional Sports) on page 10 is rewritten as “In addition to the baseball sports, studies on salary discrimination have also been conducted in other sports [2,15,16,17,18,19, 20].” Regarding salary discrimination in football, Swift analyzed the data of 1,397 MLS players……“Simmons summaries the research findings of papers published in Journal of Sports Economics on the workings of professional sports labor markets. In sum, the evidence of hiring and exit discrimination in various leagues looks stronger than the evidence for salary discrimination. [21].”

-> Carry out a comprehensive proofreading of the whole manuscript.

Response: Thank you for the valuable suggestion.

-> Cross-check the consistency of the citation with the references.

Response: The cited article has been updated.

The paragraph from the last line on page 8 to page 9 is rewritten as “Christiano analyzed the game data of 203 players in 1977 and 356 players in 1987 in terms of eight variables including the number of baseball seasons before 1987, batting average in the last year's season, number of home runs in the last year, retransmission royalty, guard position, left or right-handed batter, playoff or not, and racial origin. The results of multivariate regression analysis showed that inter-racial salary discrimination existed among the baseball teams in 1977, but the impact of racial origin on player salaries was not significant in 1987, indicating that racial origin had a different impact on player salaries ten years later. Subsequent studies also focused on inter-racial salary discrimination [12].”

The original citation is “[12]Christiano, K. J. Salary discrimination in Major League Baseball: The effect of race. Sociology of Sport Journal. 1986; 3(2): 144-53”.

We updated it as “Christiano, K. J. Salaries and race in professional baseball: Discrimination 10 years later. Sociology of Sport Journal. 1988; 5(2): 136-149.”

-> Impacts word may be replaced with impact

Response: The Manuscript has been updated.

- Title is rewritten as “The Impact of National Culture, Altruism, and Risk Preference on Salaries: The Case of the Major League Baseball”

-> The literature review may be structured better with sub-topics on different variables of study

Response: The literature Review has been added sub-topics.

 2.1: Salary Discrimination in Baseball

 2.2: Salary Discrimination in Other Professional Sports

 2.3: Salary Discrimination in Asian Professional Sports

 2.4: Impact of Cultural Indicators on Salary

-> The authors have not generated hypotheses, it is important to do so

Response: The Manuscript has been updated.

- page 45 Endnotes 3 is rewritten as “The larger the cultural distance with domestic players is, the greater the salary premium for the foreign player is.” 

-> Add a research model for clarity

Response: Thank you for the valuable suggestion.

1. The first paragraph on page 16 is rewritten as “Cultural indicators provide a new perspective for the studies of salary discrimination in professional sports. Based on related player data (e.g., salaries, nationality, personal characteristics, and performance) and using the salary equation of Mincer and methodology of Jane, we constructed a model to examine the impact of nationality and CD on player salaries[5,33]. 

〖logSal〗_it=α+u_i+β_1 〖Nat〗_i+β_2 CD_i+β_3 P_it+β_4 C_it+v_it+ε_it (1)”

2. The first paragraph on page 17 is rewritten as “Equation (2) is the first method for measuring the CD, and is based on the study of Hofstede [4], involving six cultural dimensions. PDI denotes the degree to which organizational or social members accept the unequal distribution of power, namely, the degree to which people in a cultural context accept authority or privilege; IDV/COL denotes whether a society overall cares more about individual or collective interests; UAI denotes the degree to which a society can overall tolerate uncertainty; MAS/FEM denotes the degree to which masculinity is dominant in a cultural context; LTO/STO denotes the trade-off between long-term and short-term interests in a community; and IVR denotes the degree to which social members intend to restrain their desires. 

〖CD〗_i=∑_(k=1)^6▒{(I_kj-I_kt )^2/V_k } /6. (2)

where, 〖CD〗_i denotes the Hofstede CD of the i-th player between his/her country of birth (j) and the USA (t); I_kj denotes the k-th cultural dimension indicator of a player in his/her country of birth (j); I_kt denotes the k-th cultural dimension indicator of the USA (t); and V_k denotes the variance of the k-th cultural dimension indicator. 3 ”

3. On page 18 is rewritten as “ 〖CD〗_i=|z_kj-z_kt |. (3)

The second equation for measuring the CD (i.e., Equation (3)) is based on the GPS data [8]. 〖CD〗_i denotes the GPS-based CD of the i-th player between his/her country of birth (j) and the USA (t); z_kj denotes the preference of the country of birth (j) in the k-th GPS indicator; and z_kt denotes the preference of the USA (t) in the k-th GPS indicator. ”

-> The tables may be made more concise and presented better, some may be removed and added in the appendix section

Response: The Table 5 & Table 6 has been updated. The complete result will be provided by author if needed.

On page 29 is rewritten as“

Table 5 Regression analysis of pitchers’ risk preference (uncertainty avoidance) and altruism (collectivism) by RE models

Dependent variables: logsal

VARIABLES Model 1 Model 2 Model 3 Model 4 Model 5 Model 6 Model 7 Model 8 Model 9 Model 10

Player characteristics

RIS/UAI 1.88*** 1.05** 1.84*** 1.06*** 1.88*** 0.012*** 0.0099*** 0.011*** 0.0095*** 0.011***

 (0.33) (0.41) (0.32) (0.41) (0.32) (0.0021) (0.0027) (0.0021) (0.0027) (0.0021)

ALT /COL -0.42*** -0.024 -0.44*** -0.063 -0.46*** -0.0018** 0.00016 -0.0017** 0.00016 -0.0017**

 (0.13) (0.17) (0.13) (0.17) (0.13) (0.00080) (0.0010) (0.00080) (0.0010) (0.00079)

Control vble. a YES YES YES YES YES YES YES YES YES YES

League YES YES YES YES YES YES

Team YES YES YES YES YES YES

Year YES YES YES YES YES YES

Constant 12.4*** 8.58*** 12.5*** 8.13*** 12.2*** 12.9*** 7.29*** 13.1*** 7.03*** 12.9***

 (1.54) (2.03) (1.53) (2.02) (1.52) (1.15) (1.51) (1.15) (1.50) (1.14)

Observations 2,779 2,779 2,780 2,779 2,780 3,185 3,185 3,186 3,185 3,186

Number of player_id 1,051 1,051 1,051 1,051 1,051 1,213 1,213 1,213 1,213 1,213

R-squared 0.6034 0.4038 0.6124 0.4247 0.6164 0.6585 0.4265 0.6639 0.4477 0.6697

Notes: 1. Standard errors in parentheses *** p<0.01, ** p<0.05, * p<0.1. 2. Most research in this area uses both cluster correction and a correction for heteroscedasticity. Therefore, our results using a team level cluster correction and a correction for heteroscedasticity are employed and the results support the previous findings. 3. a Other control variables include AGE, SQAGE, Tenure, SQTenure, BMI, SQBMI, CG, SHO, WIN, LOSE, HRA, INNING, BBP, and the result will be provided by author if needed. ”

On page 34 is rewritten as“

Table 6 Regression analysis of fielders’ risk preference (uncertainty avoidance) and altruism (collectivism) by RE models

Dependent variables: logsal

VARIABLES Model 1 Model 2 Model 3 Model 4 Model 5 Model 6 Model 7 Model 8 Model 9 Model 10

Player characteristics

RIS /UAI -0.71 -0.66 -0.51 -0.55 -0.54 -0.0033 -0.0029 -0.0025 -0.0027 -0.0028

 (0.58) (0.58) (0.58) (0.58) (0.57) (0.0033) (0.0033) (0.0033) (0.0033) (0.0032)

ALT /COL 0.62** 0.63** 0.56** 0.54** 0.53** 0.0030** 0.0032*** 0.0031*** 0.0028** 0.0028**

 (0.27) (0.27) (0.27) (0.27) (0.27) (0.0012) (0.0012) (0.0012) (0.0012) (0.0012)

Control vble. a YES YES YES YES YES YES YES YES YES YES

League YES YES YES YES YES YES

Team YES YES YES YES YES YES

Year YES YES YES YES YES YES

Constant 0.84 0.76 1.49 1.19 1.73 1.50 1.41 2.16 2.03 2.44

 (2.47) (2.47) (2.47) (2.47) (2.44) (2.20) (2.21) (2.19) (2.19) (2.17)

Observations 2,144 2,144 2,144 2,144 2,144 2,529 2,529 2,529 2,529 2,529

Number of player_id 764 764 764 764 764 900 900 900 900 900

R-squared 0.4186 0.4217 0.4591 0.4570 0.4676 0.4784 0.4800 0.5332 0.5368 0.5472

Notes: 1. Standard errors in parentheses *** p<0.01, ** p<0.05, * p<0.1. 2. Most research in this area uses both cluster correction and a correction for heteroscedasticity. Therefore, our results using a team level cluster correction and a correction for heteroscedasticity are employed and the results support the previous findings. 3. a Other control variables include AGE, SQAGE, BMI, SQBMI, SB, FB, AB, DB, CS, RS, and the result will be provided by author if needed. ”

-> A section on discussion and implications should be added before conclusion

Response: Thank you for the valuable suggestion.

-On page 36 is rewritten as “Many previous studies have demonstrated the race-related or nationality-related salary discrimination in professional sports [1,2,3,13,16,22], and their estimation results are compared with each other in Table A (see Appendix). Regarding the CD-related salary estimation in professional sports, Jane analyzed the salaries of Japan’s professional baseball players, finding that the salary premium of players from Holland with the largest CD (6.44) is 10.45% higher than that of players from Italy with the smallest CD (0.94) [5]. 

Moreover, other cross-cultural studies cover topics such as salary gap in multinational companies [24,25], impact of cultural background on financial and non-financial rewards [26], and professional players [5]. Many studies focused on salary gap caused by CD between company employees, but few studies have discussed the salary gap from the perspective of cross-cultural background in sports leagues in detail. In this study, we introduced related cultural indicators of players to examine salary gap caused by cultural differences between players and further estimate the salary premium of MLB players with different nationalities. As described in Table 7, players with a salary premium of not more than 19.73% are sequentially from South Korea, Panama, Japan, Saudi Arabia, and Venezuela. Players with the lowest salary premium (0.11%) are from Australia, followed by Canada, United Kingdom, South Africa, and Germany. The premium on salaries of Australian players who have the smallest CD (0.02) from the USA is 19.61% (=(3.72-0.02)*5.3%) higher than that of players from South Korea with the largest CD (3.72) from the USA.”

-> Also mention scope for future research

Response: Thank you for the valuable suggestion.

-On page 38 is rewritten as “The above analysis offsets the deficiency in studies that examined the salary discrimination in professional sports from a culture perspective. However, this study did not analyze the accurate salary gap between players with different nationalities, or estimate the salary gap using the QR method. In addition, this study merely introduced two cultural dimensions (including altruism and risk preference) of the six cultural dimensions in the GPS. Other cultural dimensions (e.g., power distance and trust) may help further explain the impact of cultural differences. Subsequent studies may verify the above conclusions and conduct an in-depth analysis considering the mutual effect of diverse complex factors.”

-> It is a very lengthy paper and may not keep the reader interested, the paper hence needs to be trimmed down

Response: Thank you for the valuable suggestion.

Reviewers' comments:

Reviewer's Responses to Questions

Comments to the Author

1. Is the manuscript technically sound, and do the data support the conclusions?

Reviewer #1: Yes

Reviewer #2: Yes

 Has the statistical analysis been performed appropriately and rigorously?

Reviewer #1: Yes

Reviewer #2: Yes

3. Have the authors made all data underlying the findings in their manuscript fully available?

Reviewer #1: Yes

Reviewer #2: No

4. Is the manuscript presented in an intelligible fashion and written in standard English?

Reviewer #1: Yes

Reviewer #2: Yes

5. Review Comments to the Author

Reviewer #1: Impacts word may be replaced with impact

The literature review may be structured better with sub-topics on different variables of study

The authors have not generated hypotheses, it is important to do so

Add a research model for clarity

The tables may be made more concise and presented better, some may be removed and added in the appendix section

A section on discussion and implications should be added before conclusion

Also mention scope for future research

It is a very lengthy paper and may not keep the reader interested, the paper hence needs to be trimmed down

Reviewer #2: The manuscript is publishable as it discusses an important research area. It also contains all the necessary components of a good research article. However, the authors should take note of the following:

o Update the literature review with more recent studies

o Carry out a comprehensive proofreading of the manuscript.

o Checking for the consistency of the citation with the references is also suggested.

Generally, to improve the quality of the manuscript, it is suggested for the authors to read a recent article on research methodology titled “Best practices in data collection and preparation: Recommendations for reviewers, editors, and authors” and integrate some of the recommendations of the authors of the article in their manuscript.

Thanks.

6. PLOS authors have the option to publish the peer review history of their article (what does this mean?). If published, this will include your full peer review and any attached files.

Do you want your identity to be public for this peer review? For information about this choice, including consent withdrawal, please see our Privacy Policy.

Reviewer #1: No

Reviewer #2: No

---

## [Decision Letter · Decision Letter 1]

16 Mar 2023

PONE-D-22-24132R1The Impacts of National Culture, Altruism, and Risk Preference on Salaries: The Case of the Major League Baseball

PLOS ONE

Dear Dr. Wang,

Thank you for submitting your manuscript to PLOS ONE. After careful consideration, we feel that it has merit but does not fully meet PLOS ONE’s publication criteria as it currently stands. Therefore, we invite you to submit a revised version of the manuscript that addresses the points raised during the review process.

We look forward to receiving your revised manuscript.

Kind regards,

Muhammad Fareed, Ph.D

Academic Editor

PLOS ONE

Journal Requirements:

**Additional Editor Comments:**

Reviewer#1

-> The paper was adjusted sufficiently in the text but figures should be reviewed. The quality of the plots is bad. Please, redo all figures to improve them.

Reviewer#2

->The discussion and implications section that should have been added before the conclusion section is still missing.

Reviewers' comments:

Reviewer's Responses to Questions

**Comments to the Author**

1. If the authors have adequately addressed your comments raised in a previous round of review and you feel that this manuscript is now acceptable for publication, you may indicate that here to bypass the “Comments to the Author” section, enter your conflict of interest statement in the “Confidential to Editor” section, and submit your "Accept" recommendation.

Reviewer #1: (No Response)

Reviewer #3: All comments have been addressed

2. Is the manuscript technically sound, and do the data support the conclusions?

Reviewer #1: Yes

Reviewer #3: Yes

3. Has the statistical analysis been performed appropriately and rigorously? 

Reviewer #1: Yes

Reviewer #3: Yes

4. Have the authors made all data underlying the findings in their manuscript fully available?

Reviewer #1: Yes

Reviewer #3: Yes

5. Is the manuscript presented in an intelligible fashion and written in standard English?

Reviewer #1: No

Reviewer #3: Yes

6. Review Comments to the Author

Reviewer #1: The discussion and implications section that should have been added before the conclusion section is still missing

Reviewer #3: The paper was adjusted sufficiently in the text but figures should be reviewed. The quality of the plots is bad. Please, redo all figures to improve them.

7. PLOS authors have the option to publish the peer review history of their article (what does this mean?). If published, this will include your full peer review and any attached files.

Reviewer #1: No

Reviewer #3: No

---

## [Author Response · Author response to Decision Letter 1]

30 Mar 2023

Response: 

We had updated the references on page 38 and listed as follows. 

1. Jane, W. J. The impact of cultural distance on salary: the case of 

Samurai Japan. Eurasian Economic Review. 2021; 11: 85-123. 

2. Asghar, F., & Asif, M. Salaries, performance and mediating effect of 

altruistic behavior: Fresh statistical evidence from the National 

Basketball Association. International Journal of Humanities, Art and 

Social Studies (IJHAS). 2018; 3: 71-9. 

3. Pappas, J. M., & Flaherty, K. E. The moderating role of 

individual-difference variables in compensation research. Journal of 

Managerial Psychology. 2006; 21(1):19-35. 

4. Williams, L.K. “Some correlates of risk taking”, Personnel Psychology. 

1965; 18(3): 297‐310. 

In table A of the Appendix on page 45, Jane (2020) is updated as Jane 

(2021). 

Additional Editor Comments:

Reviewer#1 

-> The paper was adjusted sufficiently in the text but figures should be reviewed. The quality of the 

plots is bad. Please, redo all figures to improve them. 

Response: 

We redo all figures to improve the quality of the plots. Thank you for the reminder. 

Reviewer#2 

->The discussion and implications section that should have been added before the conclusion section is 

still missing. 

Response: 

Thank you for the valuable suggestion. The discussion and implications are included 

in the text. 

1. -On page 33, the discussion is rewritten as “Many previous studies have 

demonstrated the race-related or nationality-related salary discrimination in 

professional sports [1,2,3,13,16,22], and their estimation results are compared 

with each other in Table A (see Appendix). Regarding the CD-related salary 

estimation in professional sports, Jane analyzed the salaries of Japan’s 

professional baseball players, finding that the salary premium of players from 

Holland with the largest CD (6.44) is 10.45% higher than that of players from 

Italy with the smallest CD (0.94) [5]. 

Moreover, other cross-cultural studies cover topics such as salary gap in 

multinational companies [24,25], impact of cultural background on financial and 

non-financial rewards [26], and professional players [5]. Many studies focused 

on salary gap caused by CD between company employees, but few studies have 

discussed the salary gap from the perspective of cross-cultural background in 

sports leagues in detail. In this study, we introduced related cultural indicators of 

players to examine salary gap caused by cultural differences between players and 

further estimate the salary premium of MLB players with different nationalities. 

As described in Table 7, players with a salary premium of not more than 19.73% 

are sequentially from South Korea, Panama, Japan, Saudi Arabia, and Venezuela. 

Players with the lowest salary premium (0.11%) are from Australia, followed by 

Canada, United Kingdom, South Africa, and Germany. The premium on salaries 

of Australian players who have the smallest CD (0.02) from the USA is 19.61% 

(=(3.72-0.02)*5.3%) higher than that of players from South Korea with the 

largest CD (3.72) from the USA.” 

2. -On page 35, implications are rewritten as “The above analysis offsets the 

deficiency in studies that examined the salary discrimination in professional 

sports from a culture perspective. However, this study did not analyze the 

accurate salary gap between players with different nationalities, or estimate the 

salary gap using the QR method. In addition, this study merely introduced two 

cultural dimensions (including altruism and risk preference) of the six cultural 

dimensions in the GPS. Other cultural dimensions (e.g., power distance and trust) 

may help further explain the impact of cultural differences. Subsequent studies 

may verify the above conclusions and conduct an in-depth analysis considering 

the mutual effect of diverse complex factors. ” 

Reviewers' comments: 

Reviewer's Responses to Questions 

Comments to the Author

1. If the authors have adequately addressed your comments raised in a previous round of review and 

you feel that this manuscript is now acceptable for publication, you may indicate that here to bypass the 

“Comments to the Author” section, enter your conflict of interest statement in the “Confidential to 

Editor” section, and submit your "Accept" recommendation.

Reviewer #1: (No Response) 

Reviewer #3: All comments have been addressed 

2. Is the manuscript technically sound, and do the data support the conclusions? 

The manuscript must describe a technically sound piece of scientific research with data that supports 

the conclusions. Experiments must have been conducted rigorously, with appropriate controls, 

replication, and sample sizes. The conclusions must be drawn appropriately based on the data 

presented.

Reviewer #1: Yes 

Reviewer #3: Yes 

3. Has the statistical analysis been performed appropriately and rigorously?

Reviewer #1: Yes 

Reviewer #3: Yes 

4. Have the authors made all data underlying the findings in their manuscript fully available? 

The PLOS Data policy requires authors to make all data underlying the findings described in their 

manuscript fully available without restriction, with rare exception (please refer to the Data Availability 

Statement in the manuscript PDF file). The data should be provided as part of the manuscript or its 

supporting information, or deposited to a public repository. For example, in addition to summary 

statistics, the data points behind means, medians and variance measures should be available. If there 

are restrictions on publicly sharing data—e.g. participant privacy or use of data from a third 

party—those must be specified.

Reviewer #1: Yes 

Reviewer #3: Yes 

5. Is the manuscript presented in an intelligible fashion and written in standard English? 

PLOS ONE does not copyedit accepted manuscripts, so the language in submitted articles must be 

clear, correct, and unambiguous. Any typographical or grammatical errors should be corrected at 

revision, so please note any specific errors here. 

Reviewer #1: No 

Reviewer #3: Yes 

6. Review Comments to the Author 

Please use the space provided to explain your answers to the questions above. You may also include 

additional comments for the author, including concerns about dual publication, research ethics, or 

publication ethics. (Please upload your review as an attachment if it exceeds 20,000 characters)

Reviewer #1: The discussion and implications section that should have been added before the 

conclusion section is still missing 

Reviewer #3: The paper was adjusted sufficiently in the text but figures should be reviewed. The 

quality of the plots is bad. Please, redo all figures to improve them. 

7. PLOS authors have the option to publish the peer review history of their article (what does this 

mean?). If published, this will include your full peer review and any attached files. 

Do you want your identity to be public for this peer review? For information about this choice, 

including consent withdrawal, please see our Privacy Policy.

Reviewer #1: No 

Reviewer #3: No

---

## [Editor Report · Decision Letter 2]

4 Apr 2023

The Impacts of National Culture, Altruism, and Risk Preference on Salaries: The Case of the Major League Baseball

PONE-D-22-24132R2

Dear Dr. Jye-Shyan Wang,

We’re pleased to inform you that your manuscript has been judged scientifically suitable for publication and will be formally accepted for publication once it meets all outstanding technical requirements.

Kind regards,

Muhammad Fareed, Ph.D

Academic Editor

PLOS ONE

Additional Editor Comments (optional):

Dear Author/s,

Thank you for making the corrections as per reviewers' comments.

We are delighted to inform you that your article is accepted.

Thank you.
---

## [Editor Report · Acceptance letter]

14 Apr 2023

PONE-D-22-24132R2 

The Impact of National Culture, Altruism, and Risk Preference on Salaries: The Case of the Major League Baseball 

Dear Dr. Wang:

I'm pleased to inform you that your manuscript has been deemed suitable for publication in PLOS ONE. Congratulations! Your manuscript is now with our production department. 

Kind regards, 

on behalf of

Dr. Muhammad Fareed 

Academic Editor

PLOS ONE